# HER2-Specific Targeted Toxin DARPin-LoPE: Immunogenicity and Antitumor Effect on Intraperitoneal Ovarian Cancer Xenograft Model

**DOI:** 10.3390/ijms20102399

**Published:** 2019-05-15

**Authors:** Evgeniya A. Sokolova, Olga N. Shilova, Daria V. Kiseleva, Alexey A. Schulga, Irina V. Balalaeva, Sergey M. Deyev

**Affiliations:** 1Institute of Biology and Biomedicine, Lobachevsky State University of Nizhny Novgorod, 23 Gagarin ave., Nizhny Novgorod 603950, Russia; irin-b@mail.ru; 2Laboratory of Molecular Immunology, Institute of Bioorganic Chemistry of the Russian Academy of Sciences, 16/10 Miklukho-Maklay St., Moscow 117997, Russia; olchernykh@yandex.ru (O.N.S.); darkiseleva@mail.ru (D.V.K.); schulga@gmail.com (A.A.S.); deyev@mail.ibch.ru (S.M.D.); 3Institute of Molecular Medicine, I.M. Sechenov First Moscow State Medical University, 8-2 Trubetskaya str., Moscow 119991, Russia; 4Research Nuclear Reactor Center, National Research Tomsk Polytechnic University, 30 Lenin ave., Tomsk 634050, Russia; 5Institute of Engineering Physics for Biomedicine (PhysBio), National Research Nuclear University “MEPhI”, 31 Kashirskoe Shosse, Moscow 115409, Russia

**Keywords:** targeted therapy, DARPin, *Pseudomonas* exotoxin A, immunogenicity, ovarian carcinoma, HER2, far-red fluorescent protein

## Abstract

High immunogenicity and systemic toxicity are the main obstacles limiting the clinical use of the therapeutic agents based on *Pseudomonas aeruginosa* exotoxin A. In this work, we studied the immunogenicity, general toxicity and antitumor effect of the targeted toxin DARPin-LoPE composed of HER2-specific DARPin and a low immunogenic exotoxin A fragment lacking immunodominant human B lymphocyte epitopes. The targeted toxin has been shown to effectively inhibit the growth of HER2-positive human ovarian carcinoma xenografts, while exhibiting low non-specific toxicity and side effects, such as vascular leak syndrome and liver tissue degradation, as well as low immunogenicity, as was shown by specific antibody titer. This represents prospects for its use as an agent for targeted therapy of HER2-positive tumors.

## 1. Introduction

Nowadays, the main trend in the development of anticancer therapy is enhancing the selectivity of a therapeutic effect on a tumor in order to increase the therapeutic index [1]. Therapeutic targeting of tumor cells is possible through the use of bifunctional agents—targeted toxins that are able, on the one hand, to bind selectively with tumor cells, and on the other, to eliminate them effectively [2]. Modern approaches to the creation of targeted toxins are based on combining a toxic molecule and a tumor-specific (targeting) molecule into a single genetically encoded construct of a recombinant protein [3].

The function of the targeting module in such constructs can be performed by shortened antibody fragments and non-immunoglobulin scaffold proteins. Among non-immunoglobulin scaffolds, DARPin proteins are currently considered to be the most promising, due to a number of advantages over antibodies, such as small size, high aggregative stability and ease of production in heterologous expression systems [4].

Among the different sources of toxic molecules, natural protein toxins showed better efficiency for creating such recombinant constructs. These are multidomain proteins with an enzymatic mechanism of action. One of the most effective protein toxins is exotoxin A from *Pseudomonas aeruginosa* (PE): Just a few molecules of this toxin are sufficient to kill a target cell. PE consists of three domains: Domain I selectively binds tothe α-2 macroglobulin receptor of animal cells and provides the internalization of the toxin molecule into a cell; domain II contains furin cleavage site and disulfide bonds that are reduced by protein disulfide isomerases, thus participating in the intracellular processing of the molecule; domain III has the catalytic activity, it ADP-ribosylates eukaryotic elongation factor 2, thereby blocking the protein biosynthesis in a cell, which ultimately leads to its death. The domain structure of PE makes it possible to use its shortened variants, which retain catalytic activity, while replacing its own binding domain with targeting molecule of the required specificity [5,6].

The high toxicity of PE makes it an effective component of anticancer therapy, but it also can lead to unwanted side effects with hepatotoxicity being the main one [7]. There are two basic mechanisms mediating hepatotoxicity of PE-based targeted agents: (i) The presence of the target receptor on hepatocytes and (ii) the attack of the liver by T-lymphocytes caused by overproduction of tumor necrosis factor α by Kupffer liver cells under the influence of PE [7,8]. Another serious drawback of PE arising from its bacterial origin is its high immunogenicity which significantly limits its use in clinical practice. The application of PE derivatives causes the formation of neutralizing antibodies, which reduces the effectiveness of the targeted toxin upon repeated administration and increases the risk of anaphylactic reactions. This problem is especially acute in the treatment of patients with solid tumors whose immune system functions normally [9,10]. Various approaches have been tested to solve this problem, such as PE mutagenesis followed by chemical modification (PEGylation), suppression of the patient’s immune system, detection and elimination of immunodominant epitopes of B- and T-lymphocytes by mutagenesis [11]. The last approach is the most versatile and compatible with different regimens of tumor treatment. Both immunogenicity and general toxicity should be addressed in preclinical studies of PE-based therapeutic agents.

We have previously created and characterized in detail the DARPin-PE40 targeted toxin based on 40 kDa PE fragment and HER2-specific DARPin: Its high and selective toxicity against HER2-overexpressing tumor cells realized through apoptosis induction was shown in vitro, as well as pronounced antitumor effect was revealed in vivo [12]. The HER2 protein of the human epidermal growth factor receptor family is overexpressed in many types of cancer [13,14,15]. It is involved in the pathogenesis of ovarian, breast, esophageal, stomach carcinomas, causing rapid disease progression, resistance to chemotherapy and poor prognosis for the patient. Despite a number of HER2-specific drugs approved for clinical use (trastuzumab, pertuzumab, lapatinib, trastuzumab emtansine, neratinib), successful treatment of the above mentioned HER2-positive tumors is still a serious problem, therefore the development of HER2-targeted drugs holds sway [16]. Thereby, further development of the targeted toxin DARPin-PE40 was a promising task, so we designed its version with reduced immunogenicity of the PE fragment, DARPin-LoPE [17].

In this work, we showed that DARPin-LoPE, a new version of the targeted toxin, effectively suppresses the growth of HER2-positive human ovarian carcinoma xenografts, while showing significantly less toxicity and immunogenicity than DARPin-PE40, which provides prospects for its use as an agent for targeted therapy of HER2-positive tumors.

## 2. Results

### 2.1. Targeted Toxin DARPin-LoPE and Its Cytotoxicity In Vitro

We have previously designed an agent for highly effective targeted therapy of HER2-positive tumors, the recombinant targeted toxin DARPin-LoPE (Figure 1A; the amino acid sequence is given in Appendix A). A targeting module in this molecule is a non-immunoglobulin DARPin molecule (DARPin9.29), capable of binding to the HER2 protein with high affinity (Kd = 3.8 nM) [18]. As a cytotoxic module, we used the fragment of *Pseudomonas* exotoxin A, lacking the natural receptor-binding domain Ia, domain Ib, most of domain II, as well as human B lymphocyte epitopes, and denoted by “LoPE” (molecular weight 25 kDa) [19]. We have previously shown that this targeted toxin specifically binds to HER2-overexpressing tumor cells, then internalizes and causes their apoptosis, which together elucidates its high cytotoxicity against HER2-positive target tumor cells with IC_50_ values lying in the picomolar range [17]. 

The gene construct encoding this fusion protein was expressed in *E. coli* BL21(DE3) cells. The DARPin-LoPE fusion protein was purified in two stages sequentially by metal-chelate affinity and anion-exchange chromatography, followed by removal of lipopolysaccharides by affinity chromatography on polymyxin B.

The cytotoxicity of DARPin-LoPE targeted toxin against human ovarian adenocarcinoma cells SKOV3.ip1, characterized by high expression of the HER2 protein (as shown in Appendix A), was evaluated by the MTT assay. DARPin-LoPE protein was shown to significantly reduce the viability of SKOV3.ip1 cells in the picomolar concentration range with an IC_50_ value of 100 pM (Figure 1B). At the same time, IC_50_ value for the previously created targeted toxin DARPin-PE40 containing PE40 fragment of *Pseudomonas* exotoxin A [12] was equal to 2 pM (Figure 1B). A similar difference in the cytotoxicity level of these targeted toxins was previously observed for another HER2-overexpressing cell line—SKBR-3 [12,17]. A decrease in cytotoxicity for a number of cell lines has also been shown for the similar PE-based targeted toxins of other specificities [19,20]. In addition to a number of amino acid substitutions ensuring the elimination of human B lymphocyte epitopes, the LoPE fragment structurally differs from the PE40 fragment also by the absence of domain Ib and most of domain II (except a small region containing the furin cleavage site). Despite the fact that, in general, these domains of the PE molecule were considered not to contribute to the cytotoxicity mechanism [20], some variations may occur depending on the biology of specific tumor cells, as well as the specific target on the cell surface that binds with the targeted toxin and makes it enter the cell. As significant changes in the biology of tumor cells (particularly, the HER2 functioning) are considered in three-dimensional tumor model as compared to monolayer culture, that may result in a decrease in the effectiveness of the targeted agents [21], we also evaluated the cytotoxicity of DARPin-LoPE against tumor spheroids. It was shown that the growth of human ovarian adenocarcinoma spheroids SKOV-kat is suppressed by 50% at concentrations of the targeted toxin of about 10 nM (Figure 1C).

### 2.2. General Toxicity and Immunogenicity of the Targeted Toxins DARPin-LoPE and DARPin-PE40 In Vivo

The toxicity and immunogenicity of DARPin-LoPE were evaluated in comparison with DARPin-PE40. These proteins were administered to healthy immunocompetent BALB/c mice in two courses of four doses every other day (Figure 2). DARPin-PE40 was administered at a dose of 10 µg per injection (“4 × 10”, 40 µg per course); DARPin-LoPE was administered at a similar dose of 10 µg per injection (“4 × 10”, 40 µg per course) or at a twofold dose of 20 µg per injection (“4 × 20”, 80 µg per course). The dosages were chosen according to the previous study [12], in which DARPin-PE40 has shown maximal efficiency being injected intravenously in a total amount of 80 µg. As for DARPin-LoPE, a twofold dose was also tested, since the lower in vitro toxicity of this protein had been shown, so the lower side toxicity was expected. The proteins were administered in two courses to test the possibility of repeated courses that are usually limited by protein immunogenicity. Animals were estimated by body weight, appearance and activity level. Lymphocytes-to-granulocytes ratio was monitored on days 8 and 17 after the start of each course, and the activity of alanine aminotransferase and aspartate aminotransferase in the serum was analyzed on days 6 and 18 after the start of each course. Immunogenicity of targeted toxins was assessed by the titer of antibodies specific to the protein studied, two and three weeks after the first course ended, and two and five weeks after the second course ended. Finally, animals were euthanized, and histological examination of the internal organs was performed.

#### 2.2.1. General Toxicity of DARPin-PE40 and DARPin-LoPE

The animals were estimated by body weight, appearance and activity. The first treatment course of DARPin-PE40 targeted toxin at a dose of 4 × 10 µg (40 µg per course) resulted in a significant reduction in body weight compared with the control mice and the mice treated with the DARPin-LoPE targeted toxin in both schedules (4 × 10 µg and 4 × 20 µg) (Figure 3). In addition to weight loss, the mice of this group had a ruffled fur and a decreased moving activity by the end of the first treatment course. The animals treated with DARPin-LoPE in both schedules showed no significant differences in weight from the control. After the second treatment course, none of the experimental groups showed significant differences in weight from the control group, which may be associated with the development of neutralizing antibodies that suppress the toxic effect of DARPin-PE40.

To assess whether an acute inflammation takes place, leukocytes were analyzed by flow cytometry. Analysis of leukocytes immediately and 10 days after the end of each treatment course did not reveal significant changes in the lymphocytes-to-granulocytes ratio in animals treated with DARPin-PE40 or DARPin-LoPE, as compared with the control animals. The proportion of granulocytes in all animals at the analyzed time point changed slightly, that indicates the absence of acute inflammation (see Appendix A for details).

The general toxicity of the targeted toxins was also assessed based on a histological analysis of organs *post mortem*. Spleens in mice treated with two courses of DARPin-LoPE in both schedules (4 × 10 µg and 4 × 20 µg per course) showed an increased volume of lymphatic follicles. However, in mice treated with 4 × 10 µg DARPin-PE40 per course, the hyperplasia of the white pulp was more pronounced both in the form of individual follicles and in the area of periarteriolar lymphatic sheaths, indicating a stronger immune response (Figure 4). In the lungs of mice treated with 4 × 10 µg DARPin-PE40 per course, there was a slight expansion of the peribronchovascular space. This may be due to the increased permeability of the blood vessels (vascular leak syndrome), which has previously been described as a side effect of using protein toxins based on PE40 [22]. The DARPin-LoPE protein produced a similar effect only at a twofold dose (4 × 20 µg per course) (Figure 4). In the kidneys of mice treated with 4 × 10 µg DARPin-PE40 per course, significant hemorrhage and necrosis foci were found in the cortex and medulla, and irregular blood filling was observed in the medulla blood capillaries. At the same time, the targeted toxin DARPin-LoPE administered in the same dosage did not affect the kidney morphology. Degenerative lesions of kidneys were detected only in animals treated with a twofold dose of DARPin-LoPE (4 × 20 µg per course), but they were even less pronounced (small hemorrhage foci in the cortex were observed) (Figure 4). Pathomorphological study of liver and heart did not reveal macroscopic and microscopic changes in mice treated with both targeted toxins (Figure 4).

For confirmation, the ALT and AST activities were measured in sera. These enzymes work in the liver (ALT) or in the liver, heart and skeletal muscles (AST). Normally, their activity in serum is very low, but it increases with the hepatocytes and cardiomyocytes destruction and may indicate damage to the internal organs. The ALT and AST activities were measured on days 6 and 18 after the start of each treatment course: These parameters showed the values within the physiologically normal range in both treated and control mice (as shown in Appendix A).

Thus, neither DARPin-PE40 nor DARPin-LoPE showed marked cardiotoxicity and hepatotoxicity. However, DARPin-PE40 caused changes in the immune compartment of the spleen, which is consistent with the assumption of its increased immunogenicity. Both DARPin-PE40 and DARPin-LoPE led to changes in the lungs and kidneys, but the DARPin-PE40 toxic effect was higher than that of DARPin-LoPE even with a twofold dosage of the latter.

#### 2.2.2. Immunogenicity of DARPin-PE40 and DARPin-LoPE

The immunogenicity of the proteins was assessed by the titer of specific antibodies in the serum of mice (Figure 5, Appendix A). In 20 days after the first treatment course started, a significant increase in the antibody titer was observed in all experimental groups as compared with control mice treated with PBS. The antibody titer against DARPin-LoPE was 1:10,000 and 1:30,000 in the groups of mice treated with 4 × 10 μg and 4 × 20 μg DARPin-LoPE, respectively. The most significant increase in antibody titer was observed in the group of mice treated with 4 × 10 µg DARPin-PE40: Its value reached 1:300,000.

On day 27, the antibody titer in all groups of animals treated with targeted toxins fell to a value of 1:3500 in case of 4 × 10 μg DARPin-LoPE, and to value of 1:10,000 in cases of 4 × 20 μg DARPin-LoPE and 4 × 10 μg DARPin-PE40. In all experimental groups except 4 × 20 μg DARPin-LoPE the decrease in antibody titer on day 27 was significant as compared to that on day 20. In 20 days after the second treatment course started (day 60 of the experiment), the antibody titer increased significantly in all experimental groups, compared with mice treated with PBS. In mice treated with DARPin-LoPE in both doses, it was equal to 1:300,000, and in mice treated with 4 × 10 µg DARPin-PE40 the titer value was 1:900,000. In 40 days after the second treatment course started (day 80 of the experiment), the minimum antibody titer was observed in mice treated with 4 × 10 μg DARPin-LoPE (1:10,000), slightly higher titer was in mice treated with 4 × 20 μg DARPin-LoPE (1:30,000), and the maximum antibody titer was revealed in mice treated with 4 × 10 μg DARPin-PE40 (1:300,000). In all experimental groups the decrease in antibody titer on day 80 was significant as compared to that on day 60.

Thus, DARPin-PE40 showed higher immunogenicity compared to DARPin-LoPE, since the treatment with DARPin-PE40 led to the production of a higher antibody titer. In addition, the immune response against DARPin-PE40 had a more long-term effect compared to the case of the low immunogenic variant of the protein. In mice treated with 4 × 20 µg DARPin-LoPE per course, the immune response decreased more slowly than in mice treated with 4 × 10 µg DARPin-LoPE.

### 2.3. Effect of DARPin-LoPE against Human Ovarian Carcinoma Xenograft

The efficiency of the targeted toxin DARPin-LoPE was evaluated using a fluorescent xenograft model of disseminated human ovarian carcinoma, obtained by intraperitoneal inoculation of SKOVip-kat cells (Figure 6A) in athymic mice. The SKOVip-kat cell line was obtained by transfecting SKOV3.ip1 cells with the gene of the far-red fluorescent protein Katushka [23]. The parental cell line SKOV3.ip1, overexpressing the HER2 protein, forms disseminating tumors in athymic mice when the cells are injected intraperitoneally [24]. This situation is considered clinically relevant, since ovarian cancer, unlike hematogenously metastasizing tumors, initially spreads in the peritoneal cavity, expanding along the mesothelium and invading the tissue, or disseminating as separate cells and spheroids with a peritoneal fluid flow [25].

The high level of the HER2 expression in the SKOVip-kat cells was confirmed by flow cytofluorimetry after staining with HER2-specific antibodies conjugated to FITC (Figure 6B). The cytotoxicity of DARPin-LoPE against the SKOVip-kat cells was evaluated by the MTT assay, with IC50 value equal to 30 pM (Figure 6C), which is comparable to the cytotoxicity against the parent SKOV3.ip1 cells.

SKOVip-kat cells were inoculated intraperitoneally into athymic mice. During the week, multiple fluorescent tumor nodules were formed in the peritoneal cavity, which were visualized in vivo non-invasively by whole-body fluorescence imaging [26]. The preservation of HER2-positive status of the tumor tissue formed after inoculation of SKOVip-kat cells was confirmed by immunohistochemical analysis using HercepTest as presented in Appendix A. Thus, the obtained tumor model reproduces the clinical pattern of the HER2-positive human ovarian carcinoma development and is adequate for studying the effectiveness of the HER2-specific targeted agent.

The mice were intraperitoneally treated with five doses of 5 μg DARPin-LoPE (total dose of 5 × 5 μg per mouse) or 10 μg DARPin-LoPE (total dose of 5 × 10 μg per mouse) on days 9, 11, 13, 15, 17 (Figure 7A). In vivo imaging of SKOVip-kat fluorescent tumors has shown significantly slower development of disseminated tumor nodules in the peritoneal cavity of mice treated with a dose of 5 × 10 μg DARPin-LoPE as compared to control mice (Figure 7B). A quantitative analysis of the fluorescence intensity in the peritoneal cavity revealed that tumors in all groups of mice grew exponentially (Figure 7C). The targeted toxin DARPin-LoPE in a dose of 5 × 5 μg had a slight effect on the tumor growth, while in a dose of 5 × 10 μg it significantly inhibited the growth: A statistically significant decrease in the tumor growth rate coefficient (Figure 7D) and, correspondingly, an increase in the tumor doubling time (Figure 7E) in this group compared with the control PBS-treated group, were observed. It is interesting to note that the dose of DARPin-LoPE, which has a pronounced antitumor effect against human ovarian carcinoma in vivo, lies in the same range as the dose that inhibits the growth of the tumor spheroids in vitro. Statistically significant differences in the integral fluorescence intensity in mice treated with 5 × 10 μg DARPin-LoPE, as compared to the control mice, were observed as early as 10 days after the treatment course ended, and then persisted for a month. Tumor growth inhibition in this group of animals was 62%, compared to control animals.

## 3. Discussion

The initial idea of combining a powerful toxic molecule with an antibody specific to a surface target on tumor cell [27], gave a boost to the creation of new highly effective drugs with selective action for targeted antitumor therapy. Several factors have contributed to this process. The development of molecular oncology was accompanied by the identification and study of molecular targets of tumor cells. At the same time, advances in toxicology made it possible to choose the natural toxic molecules relying on the studied mechanism of their action on the cell. Finally, progress in the technologies of genetic engineering and biotechnology led to optimized creation of non-natural molecules with desired properties. Modern targeted antitumor toxins are recombinant proteins, in which an effector part of a natural toxin molecule is combined with a small, but highly specific targeting molecule that binds to a tumor cell with high affinity. These complex molecules have extremely selective toxicity, are fully biodegradable and can be easily produced in bacteria as fusion proteins [3,28]. Being effective themselves they are easily combined with different functional components, including ones that enhance tissue penetration [29], decrease immune response to the therapy [30] or complement toxic effect by another mechanism of action [31].

*Pseudomonas* exotoxin A is a highly toxic protein of the Gram-negative bacterium *Pseudomonas aeruginosa*. This toxin is characterized by a modular structure with almost independent functioning of individual modules, as well as inherent intracellular processing and transport of the effector (catalytic) domain inside the cell to the site of action (cytoplasm) [5,6]. Such peculiar properties cause its productive use when creating targeted toxins. Thus, a number of PE-based targeted toxins specific to various tumor cell targets were created, which showed high potential as antitumor agents in preclinical trials [12,26,32,33,34,35,36,37,38]. Several targeted toxins based on exotoxin A are undergoing Phase I clinical trials [39,40,41].

One of the serious problems for the clinical use of agents containing *Pseudomonas* exotoxin A is its high immunogenicity, which prevents repeating the course of treatment in patients with a normal immune system. In fact, this limits the range of application of such agents to hematological malignant neoplasms, accompanied by dysfunction of the immune system [39,41]. To solve this problem, Ira Pastan and colleagues carried out a series of works to reduce the immunogenicity of PE by truncation of its molecule and point mutagenesis of the remaining epitopes of mouse and human B and T lymphocytes [11]. We have used one of the variants of the C-terminal (effector) PE fragment (denoted by “LoPE”) with mutated immunodominant epitopes of human B lymphocytes [19] to create a targeted toxin specific to the HER2 protein, DARPin-LoPE [17].

The main objective of this work was to compare the general toxicity and immunogenicity of the targeted toxin DARPin-LoPE with the previously created analogous targeted toxin DARPin-PE40, which showed high activity against HER2-positive tumor cells both in vitro and in vivo [12], but contained an immunogenic fragment of PE (aa 252–612, denoted by “PE40”). We have shown that in used doses, DARPin-LoPE has less non-specific toxicity and leads to less pronounced side effects, such as vascular leak syndrome, liver and kidney tissue degradation, which are known problems that limit the use of PE-based toxins in clinical practice [7,42]. Absence of weight loss during the course of DARPin-LoPE injections, as compared with DARPin-PE40, also testifies to lower side toxicity of the DARPin-LoPE toxin. DARPin-LoPE resulted in the formation of the lower antibody titer after the first treatment course, and the concentration of specific antibodies fell faster, which indicates the less efficient formation of memory cells. This study was conducted on mice; however, it was shown that the epitopes of mouse and human B lymphocytes in the PE domain III partially overlap [19], which gives grounds for predicting immunogenicity in humans as well. Thus, DARPin-LoPE seems to be a more promising agent, as compared to DARPin-PE40, in terms of non-specific toxicity and the possibility of using in repeated courses. It is worth noting that in [43] progress was also achieved in reducing the immunogenicity and toxicity of the HER2-specific PE-based immunotoxin through the use of a mutant variant of the catalytic PE domain, devoid of human B lymphocyte epitopes.

A xenograft tumor model in an immunodeficient animal does not allow evaluation and comparison of the efficiency of such agents with varying degrees of immunogenicity, since the latter depends mainly on the functioning of the immune system. However, we basically showed the antitumor efficacy of DARPin-LoPE targeted toxin against human HER2-overexpressing ovarian carcinoma xenograft SKOVip-kat: Even a single-course treatment with the targeted toxin in a dose of 50 μg caused a significant inhibition of the growth of model tumors. Thus, we conclude that the DARPin-LoPE targeted toxin is potentially effective for the treatment of HER2-overexpressing tumors, since it has shown significant in vivo antitumor activity. Nevertheless, DARPin-LoPE still can cause side toxicity and production of toxin-specific antibodies after being injected intravenously, though these unwanted effects are less pronounced, than in case of DARPin-PE40. It should be noted separately that standard therapy of ovarian carcinoma usually includes surgical removal of the tumor and subsequent chemotherapy with platinum or taxane agents [44]. However, only temporal sensitivity to chemotherapy is characteristic of ovarian cancer, which makes the probability of a successful cure only 30%. In addition, the clinical picture of this disease is characterized by multiple metastases disseminated in the peritoneal cavity, which represent the main cause of the rapid progression of the disease and high mortality [25]. The shown efficacy of the HER2-specific targeted toxin DARPin-LoPE namely against intraperitoneal metastases modeled in this study represents a significant result in the perspective of the clinical use of this agent locally.

## 4. Materials and Methods

### 4.1. Expression and Purification of Targeted Toxins DARPin-PE40 and DARPin-LoPE

Recombinant proteins DARPin-PE40 [12] and DARPin-LoPE [17] were produced in *Escherichia coli* BL21(DE3) strain cells transformed with the plasmids pDARP-PE40 and pDARP-LoPE, respectively. Purification of both proteins was performed sequentially using metal-chelate affinity chromatography on Ni^2+^-NTA column (GE Healthcare, Chicago, IL, USA) and anion-exchange chromatography on MonoQ5/50 GL column (GE Healthcare) with further purification from bacterial lipopolysaccharide using Detoxi-Gel Endotoxin Removing column (Thermo Fisher Scientific, Waltham, MA, USA) according to the manufacturer’s instructions. Expression and purification procedures are described in detail in Appendix B. The SDS-PAGE analysis was performed according to standard protocols using 12% polyacrylamide gels.

### 4.2. Cell Lines

Cell lines SKOV3.ip1 [24] and SKOV-kat [26] were cultured in McCoy’s 5A medium with 10% (*v*/*v*) fetal calf serum (HyClone) and 2 mM l-glutamine. Cells were grown in 5% CO_2_ at 37 °C. For passaging cells were carefully detached using Versene solution (PanEco, Moscow, Russia) in order to prevent proteolysis of membrane proteins. The level of HER2 expression in these cells was estimated by flow cytometry after staining of cell suspensions with HER2-specific mouse monoclonal antibodies conjugated to FITC (for details of the procedure see Appendix C).

### 4.3. Generation of a Fluorescent Human Ovarian Carcinoma Cell Line SKOVip-Kat Overexpressing HER2

To obtain a fluorescent tumor cell line, 7 × 10^4^ SKOV3.ip1 cells were transfected with mammalian expression vector pTurboFP635-N (Evrogen, Moscow, Russia) encoding the far-red fluorescent protein Katushka (commercial name TurboFP635) using Lipofectamin 2000 transfection reagent (Thermo Fisher Scientific, Waltham, MA, USA) according to the previously published protocol [45] with modifications. Cells were selected in the medium containing 2 g/L of G418 (Sigma-Aldrich, St. Louis, MO, USA) and expanded for several passages.Then 3 × 10^4^ cells with the highest fluorescence signal were sterile sorted on a FACS Aria III cell sorter (Becton Dickinson, Franklin Lakes, NJ, USA) using a yellow 561 nm laser and 610/20 channel to excite and collect TurboFP635 fluorescence, respectively. Cells were then expanded in cell culture, and 3 × 10^4^ cells were sorted again to obtain the brightest fluorescent protein subset, with further single-cell sorting to generate a monoclonal cell line. The obtained cell line named SKOVip-kat was maintained in the same way as parental cells SKOV3.ip1.

The level of HER2 expression in SKOVip-kat cells was estimated by flow cytometry after staining of cell suspensions with HER2-specific mouse monoclonal antibodies conjugated to FITC (for details of the procedure see Appendix C).

### 4.4. Confocal Microscopy

SKOVip-kat cells were seeded in 35 mm glass bottom Petri dishes, 10^5^ cells per dish, and grown overnight. Cells were then visualized using an inverted laser scanning confocal fluorescence microscope AxioObserver LSM 710 (Carl Zeiss, Oberkochen, Germany). The images were obtained with a C-Apochromat 63×/1.2 water immersion objective. Fluorescence of TurboFP635 protein was excited by the HeNe laser at 594 nm and collected in the range of 611–740 nm.

### 4.5. Cell Viability Assay

For the cytotoxicity study on monolayer culture, SKOV3.ip1 and SKOVip-kat cells were seeded in a 96-well plate (Corning, New York, NY, USA), 2000 cells per well, and grown overnight. The medium was then exchanged with the fresh one containing different concentrations of DARPin-PE40 or DARPin-LoPE targeted toxin (10^−7^–10^2^ nM) and the cells were incubated for 72 h. Cell viability was estimated using MTT assay [46]. The medium was exchanged with the fresh serum-free medium containing 0.5 mg/mL MTT (3-(4,5-dimethylthiazol-2-yl)-2,5-diphenyltetrazolium bromide, Alfa Aesar, Haverhill, MA, USA) followed by incubation for 1 h. Formazan formed from the reduction of MTT by mitochondrial dehydrogenases was dissolved in dimethyl sulfoxide (PanEco, Moscow, Russia), and the absorbance was measured spectrophotometrically at 570 nm with Synergy MX plate reader (BioTek, Winooski, VT, USA). Cell viability was calculated as a ratio of the optical density of treated to one of untreated cells (given as a percentage).

For the cytotoxicity study on spheroid culture, SKOV-kat cells were seeded in a 96-well round bottom ultra-low attachment plate (Corning), 1000 cells per well, and grown for three days until compact cell aggregates (spheroids) formed. Spheroids were then incubated with different concentrations of DARPin-LoPE for 96 h. Spheroids were visualized by phase contrast microscopy using an Axiovert 200 inverted microscope with an EC Plan-Neofluar 10×/0.3 objective (Carl Zeiss, Oberkochen, Germany). Spheroid volume (V, mm^3^) was calculated with the following formula:V = a × b^2^/2,
where a and b are the larger and the smaller spheroid diameters (µm), respectively.

Cytotoxicity of DARPin-LoPE against spheroids was evaluated according to spheroid volume on the final day of incubation in the presence of the targeted toxin: Relative cell viability was in this case calculated as a percentage of the mean volumes of treated to untreated spheroids.

Data analysis was performed using GraphPad Prism 6 software. IC_50_ was calculated by nonlinear regression using the four-parameter dose-response model.

### 4.6. Animals

Six- to eight-week-old female BALB/c and BALB/c-nude athymic mice were purchased from the specific-pathogen-free licensed nursery of Shemyakin and Ovchinnikov Institute of Bioorganic Chemistry of the Russian Academy of Sciences. Animals were kept in well-ventilated polypropylene cages with a 12-h light-dark cycle, fed with sterilized standard laboratory food and supplied with water ad libitum. All experimental procedures were approved by the Animal Care and Use Committee of the Institute of Biology and Biomedicine of Lobachevsky State University of Nizhny Novgorod (ethical project code №4, 3 April 2017). The mice were randomly assigned to experimental groups 5–8 animals each and kept in a cage-controlled manner so that mice from different experimental groups were kept together.

### 4.7. Study of General Toxicity and Immunogenicity of the Targeted Toxins

To study the general toxicity and immunogenicity of the DARPin-PE40 and DARPin-LoPE targeted toxins, these proteins were administered to healthy immunocompetent BALB/c mice. Five animals were randomly assigned to each group and received two-course treatment according to three treatment schedules: Four doses of 10 µg DARPin-PE40 every other day (4 × 10, 40 µg per course, 80 µg total dose), four doses of 10 µg DARPin-LoPE every other day (4 × 10, 40 µg per course, 80 µg total dose), or four doses of 20 µg DARPin-LoPE every other day (4 × 20, 80 µg per course, 160 µg total dose). Proteins were injected in a volume of 100 μL in phosphate-buffered saline (PBS, PanEco) into the tail vein, the control group received injections of PBS.

During the course of injections, the animals were daily controlled for their appearance, activity and body weight; between the courses, body weight was measured every fifth day. Blood sampling was performed from the facial vein of animals. The study schedule included no more than one blood sampling per day. All invasive manipulations were performed under general anesthesia—1.8 mg of Zoletil (Virbac, Carros, France) and 32 mg of xylazine hydrochloride (Rometar, Bioveta, Czech Republic) were injected intraperitoneally in 100 μL of PBS.

To study the general toxicity, the blood cell count and transaminase activity were monitored during the experiment, and afterwards histological analysis of organs was performed. Analysis of leukocytes in the blood of animals was carried out on day 0 (before the start of injections), then on days 8 and 17 after each course of injections started (i.e., on days 8, 17, 47 and 56 of the experiment). To this end, 20 μL of whole blood was mixed with heparin solution containing 50 units of activity (Sintez, Moscow, Russia), and then 500 μL of lysis buffer was added and incubated for 10 min on ice. After that, 600 μL of PBS was added and the mixture was centrifuged at 100× *g* for 10 min at +4 °C. The procedure of lysis and centrifugation was repeated, and finally cells were resuspended in 100 μL of PBS with 1% bovine serum albumin. The obtained suspension of leukocytes was analyzed on a NovoCyte flow cytometer (ACEA Biosciences, San Diego, CA, USA). Statistical analysis was performed using Student’s criterion with Bonferroni correction.

Analysis of the activity of alanine aminotransferase (ALT) and aspartate transaminase (AST) in the serum of mice was performed on day 0 (before the start of injections), then on days 6 and 18 after each course of injections started (i.e., on days 6, 18, 45 and 57 of the experiment). The enzyme activity was measured with the use of colorimetric test kits by Reitman-Frankel method (Olvex Diagnosticum, Saint Petersburg, Russia) in accordance with the manufacturer’s protocol. The optical density of the reaction medium, which is proportional to the activity of transaminases, was measured spectrophotometrically at 560 nm with Infinite M1000 Pro spectrophotometer (Tecan, Männedorf, Switzerland). The time points for leukocyte count and ALT/AST activity analysis were chosen so that both acute and delayed toxicity could be evaluated, and the tests were done at different time points, as both large blood sample volumes and frequent sampling would contribute to mice health state. The maximal blood volume and sampling frequency that does not affect the chosen parameters were estimated in the preliminary experiment.

For the histological analysis spleen, liver, heart, lungs and kidneys were taken after euthanasia of animals on the 90th day of the experiment. The organs were fixed with 4% formaldehyde solution in PBS, the pieces of the fixed organs were dehydrated in a series of ethanol solutions of increasing concentrations and embedded in paraffin. Histological sections were made on a rotary microtome RM2255 (Leica Biosystems, Wetzlar, Germany) with a slice thickness of 3 microns. Sections were hydrated in ethanol solutions with decreasing concentration, stained with Meyer’s hematoxylin (PanEco, Moscow, Russia) for 3 min and eosin (PanEco) for 2 min, dehydrated in ethanol and covered with glass in Acrytol mounting medium (Leica Microsystems, Wetzlar, Germany). The images were obtained using DMI6000B optical inverted microscope (Leica Microsystems) with magnification ×100, ×200, ×400.

To study immunogenicity, the titer of antibodies specific to the administered proteins was analyzed. The antibody titer was estimated with ELISA on day 0 (before the start of injections of the targeted toxins), and then on days 20, 27, 60 and 80 of the experiment. The ELISA procedure is described in Appendix D.

### 4.8. Study of Antitumor Effect of DARPin-LoPE in Fluorescent Xenograft Models

To estimate the antitumor effect of DARPin-LoPE, fluorescent human ovarian carcinoma xenografts were obtained by inoculation of 4 × 10^6^ SKOVip-kat cells in 200 µL PBS intraperitoneally in BALB/c-nude athymic mice. The HER2 expression in the developing tumors was analyzed ex vivo by HercepTest (Dako, Glostrup, Denmark) according to the manufacturer’s instructions. SKOVip-kat fluorescent tumors were visualized by the whole-body in vivo imaging using a home-built back-reflectance imaging system (Institute of Applied Physics RAS, Nizhny Novgorod, Russia) [47]. Fluorescence was excited by a narrow-band light-emitting diode at 585 nm; for the emission collection, band-pass filter 628–672 nm was used. To estimate the tumor growth, fluorescence imaging was performed 2–3 times per week for several weeks. Obtained fluorescence images were analyzed using ImageJ software (National Institute of Health, Bethesda, MD, USA). The integral fluorescence intensity calculated in the peritoneal area at each time point (F) was used as a quantitative measure of tumor growth. Nonlinear regression of the obtained data was performed using GraphPad Prism 6 (GraphPad Software) using the exponential growth equation:
F = F_0_ × e^kt^,
where F_0_ is the integral fluorescence in the abdominal cavity at the initial moment of time, k is the tumor growth rate coefficient.

Eight days after the inoculation of the cells, when tumor integral fluorescence reached ∼1 × 10^6^ a.u., eight animals were randomly assigned to each treatment group and received intraperitoneal injections of DARPin-LoPE according to two treatment schedules: Five doses of 5 µg (5 × 5 µg) or five doses of 10 µg (5 × 10 µg) in 200 µL PBS every other day (i.e., on days 9, 11, 13, 15 and 17 after tumor cells inoculation). In the control group, mice were injected with 200 µL PBS every time.

To calculate tumor growth inhibition coefficient (TGI), the following formula was used:
TGI (%) = [(F_control_ − F_treatment_) × 100%]/F_control_
where F is an integral fluorescence intensity in the peritoneum area at a selected time point.

Statistical analysis was performed using one-way ANOVA and Dunnett’s test (GraphPad Prism 6 software).

## 5. Conclusions

In this work, we showed that the targeted toxin DARPin-LoPE composed of HER2-specific DARPin and low immunogenic exotoxin A fragment effectively inhibits the growth of HER2-expressing human ovarian carcinoma xenografts, while showing significantly less toxicity and immunogenicity than its previous version DARPin-PE40. These results present prospects for DARPin-LoPE use as an agent for targeted therapy of HER2-positive tumors.

## Figures and Tables

**Figure 1 ijms-20-02399-f001:**
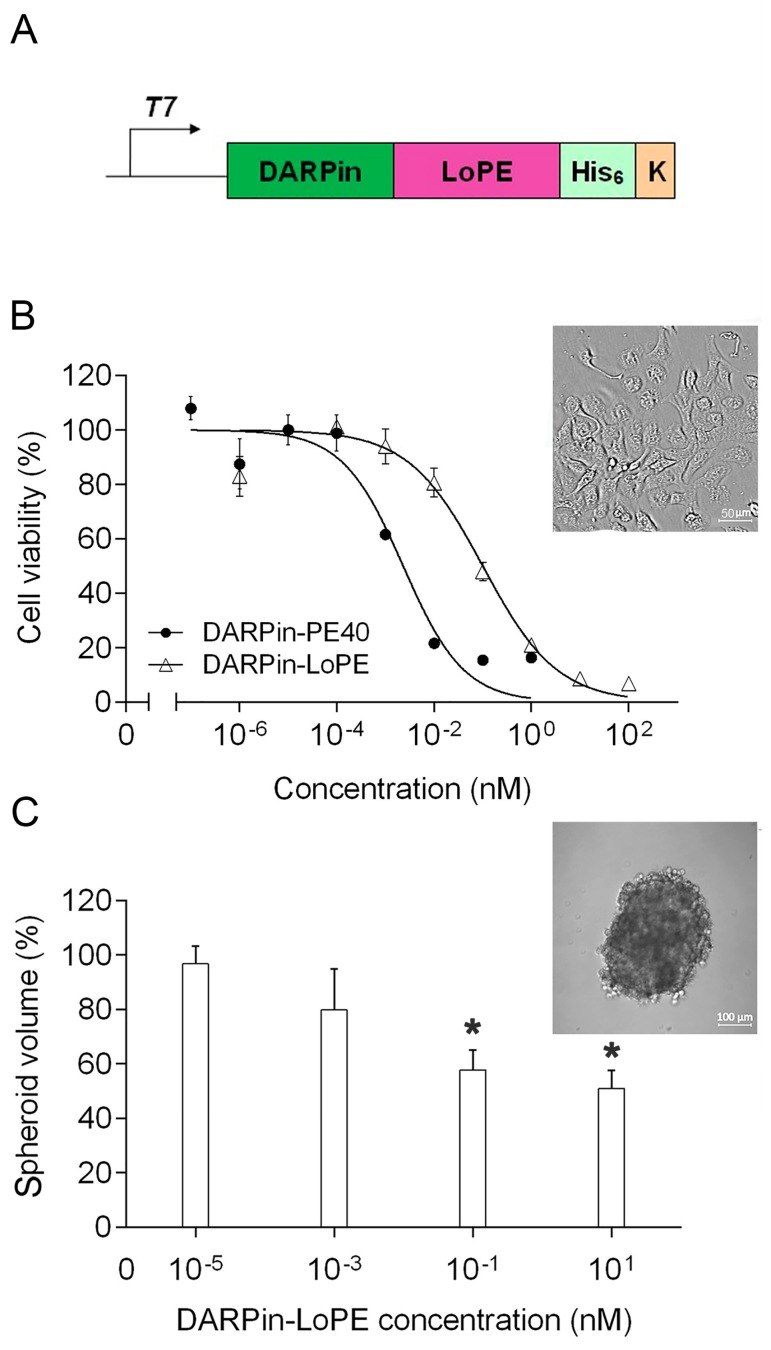
Targeted toxin DARPin-LoPE and its cytotoxicity against human carcinoma 2D and 3D models in vitro. (**A**) Gene construct encoding DARPin-LoPE. DARPin (dark green), HER2-specific DARPin9.29; LoPE (purple), truncated *Pseudomonas* exotoxin A deprived of domains Ia, Ib, most of domain II and human B cells epitopes; His6 (light green), C-terminal hexahistidine tag; K (orange), KDEL sequence. The fusion gene is under control of the T7 promoter; (**B**) Relative viability of the HER2-positive human ovarian adenocarcinoma cells SKOV3.ip1 in 2D monolayer culture (MTT assay) after a 72 h treatment with different concentrations of DARPin-PE40 (black circles) or DARPin-LoPE (transparent triangles); (**C**) Relative volume of 3D spheroids of the HER2-positive human ovarian adenocarcinoma cells SKOV-kat after a 96 h treatment with different concentrations of DARPin-LoPE. The data are represented as mean ± SEM. “*” indicates a value that significantly differs from the respective control value at *p* < 0.05 (Dunnett’s test, *n* = 6).

**Figure 2 ijms-20-02399-f002:**
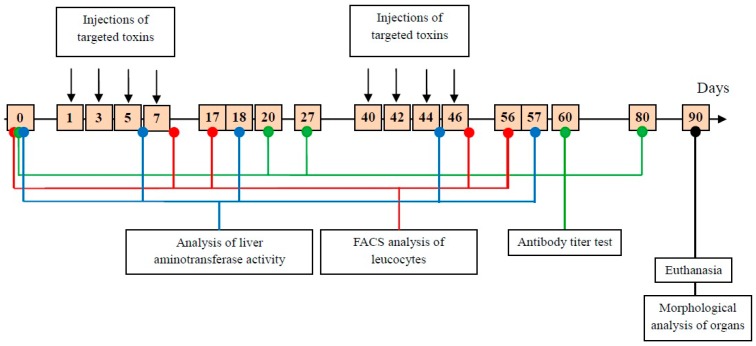
Scheme of the study of DARPin-LoPE and DARPin-PE40 general toxicity and immunogenicity. “0” denotes the analysis of the parameters studied in animals before the treatment course started. The day of the first injection of the targeted toxins was set as day 1. Treatment with the targeted toxins was performed in two courses of four injections every other day, with a break of one month between courses, i.e., on days 1, 3, 5, 7 (first course) and 40, 42, 44 and 46 (second course). The leucocytes were analyzed by flow cytometry on days 8 and 17 after the start of each course (i.e., on days 8, 17, 47 and 56, marked in red). The activity of alanine aminotransferase (ALT) and aspartate aminotransferase (AST) in the serum was measured on days 6 and 18 after the start of each course (i.e., on days 6, 18, 45 and 57, marked in blue). The antibody titer test was performed two and three weeks after the first treatment course ended, and two and five weeks after the second treatment course ended (i.e., on days 20, 27, 60 and 80, marked in green). Animals were euthanized on day 90, and organs were taken for histological study (marked in black).

**Figure 3 ijms-20-02399-f003:**
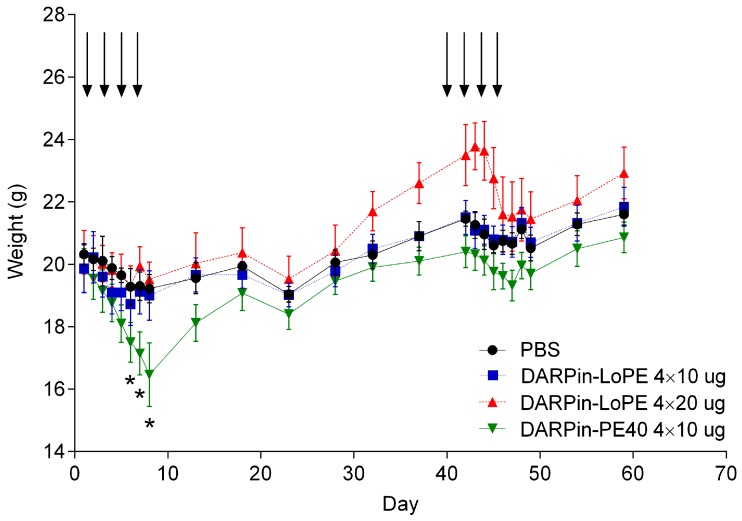
Mean weight of mice in different groups. The days of the targeted toxins injections are indicated with arrows (4 injections per course, 2 courses). PBS, mice treated with phosphate-buffered saline (control group); DARPin-LoPE 4 × 10 µg, mice treated with 10 µg DARPin-LoPE for 4 doses every other day (40 µg per course, 80 µg total); DARPin-LoPE 4 × 20 µg, mice treated with 20 µg DARPin-LoPE for 4 doses every other day (80 µg per course, 160 µg total); DARPin-PE40 4 × 10 µg, mice treated with 10 µg DARPin-PE40 for 4 doses every other day (40 µg per course, 80 µg total). The data are represented as mean ± SEM. “*” indicates a value that significantly differs from the respective control value at *p* < 0.05 (Student’s test with Bonferroni correction, *n* = 5).

**Figure 4 ijms-20-02399-f004:**
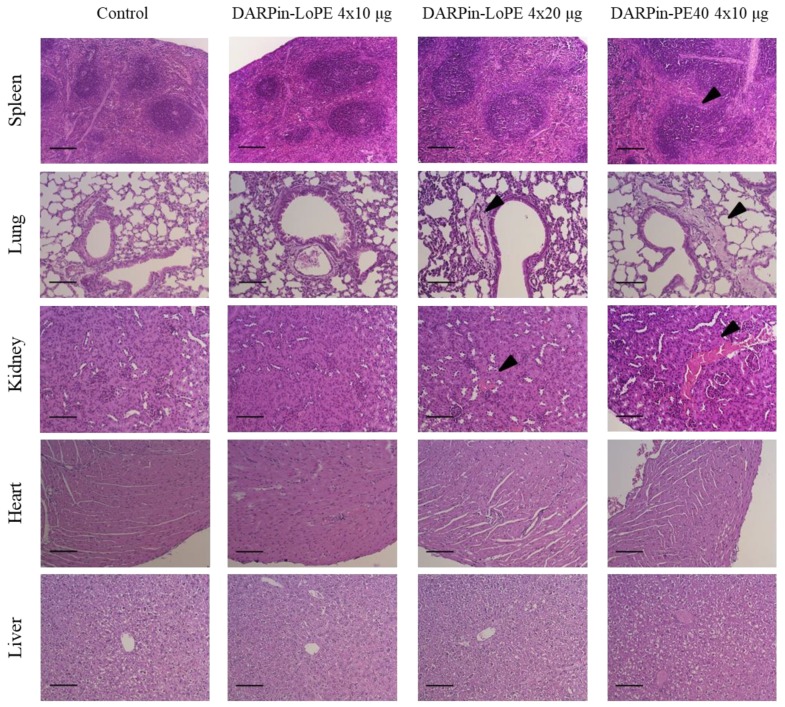
Hematoxilin and eosin staining of organ sections of mice in different groups. Degenerative tissue lesions upon treatment with the targeted toxins are indicated with arrows. In mice treated with 4 × 10 μg DARPin-PE40 the portion of the white pulp of spleen is increased in a zone of periarteriolar sheaths. In mice treated with 4 × 10 μg DARPin-PE40 or 4 × 20 μg DARPin-LoPE the peribronchovascular space is dilatated in the lungs, as well as hemorrhage and necrosis sites are observed in kidneys. For spleen images scale bar is 200 µm, magnification ×100; for lung, kidney, heart and liver scale bar is 100 µm, magnification ×200.

**Figure 5 ijms-20-02399-f005:**
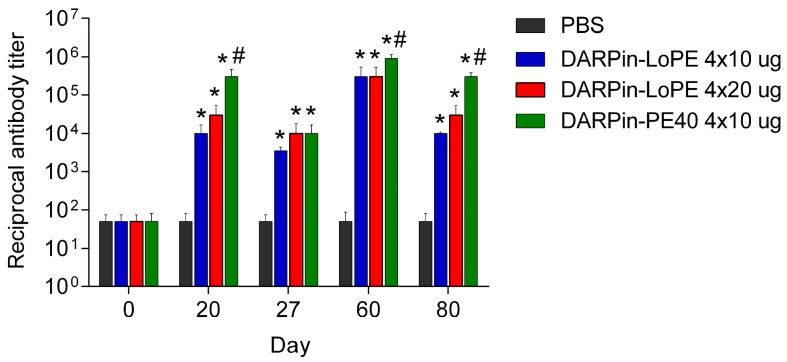
Dynamics of titers of antibodies specific to the targeted toxins. The day of the first injection of the targeted toxins was set as day 1. 0 is for the measurements of antibody titers in mice serum before the first treatment course started. PBS, mice treated with phosphate-buffered saline (control group); DARPin-LoPE 4 × 10 µg, mice treated with 10 µg DARPin-LoPE for 4 doses every other day (40 µg per course, 80 µg total); DARPin-LoPE 4 × 20 µg, mice treated with 20 µg DARPin-LoPE for 4 doses every other day (80 µg per course, 160 µg total); DARPin-PE40 4 × 10 µg, mice treated with 10 µg DARPin-PE40 for 4 doses every other day (40 µg per course, 80 µg total).“*” indicates a value that significantly differs from the respective control (PBS) value at *p* < 0.05; “#” indicates a value that significantly differs from the respective values in groups DARPin-LoPE 4 × 10 µg and DARPin-LoPE 4 × 20 µg at *p* < 0.05 (Student’s test with Bonferroni correction, *n* = 5). The data are represented as mean ± SEM.

**Figure 6 ijms-20-02399-f006:**
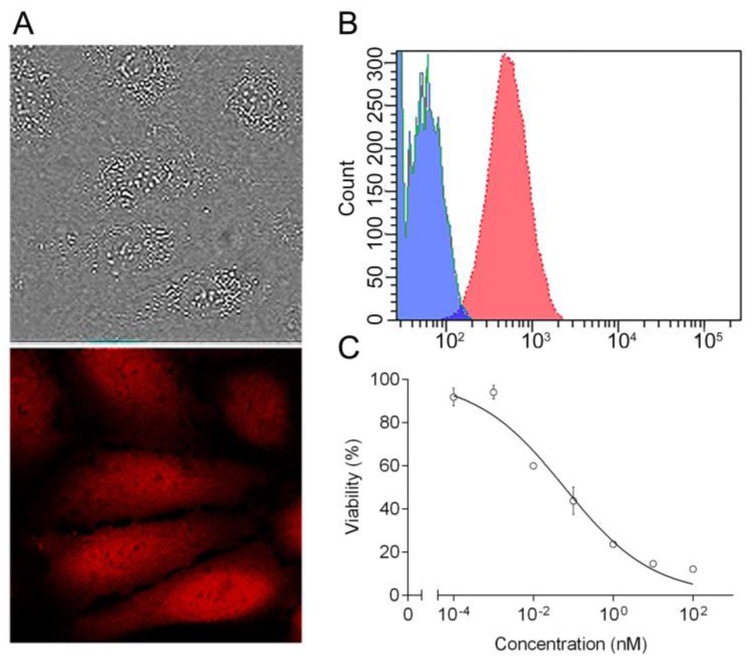
Fluorescent human ovarian carcinoma cells SKOVip-kat and their response to DARPin-LoPE treatment in vitro. (**A**) Visualization of SKOVip-kat cells expressing protein Katushka (red) in transmitted light (on the top) and by confocal microscopy (on the bottom). Image size 135 × 135 µm; (**B**) Analysis of surface content of the HER2 protein on SKOVip-kat cells stained with FITC-labeled anti-HER2 antibody (red filling) or with FITC-labeled isotypic control (blue filling) and analyzed by flow cytometry; (**C**) Relative viability of SKOVip-kat cells in 2D monolayer culture (MTT assay) after a 72 h treatment with different concentrations of DARPin-LoPE. The data are represented as mean ± SEM.

**Figure 7 ijms-20-02399-f007:**
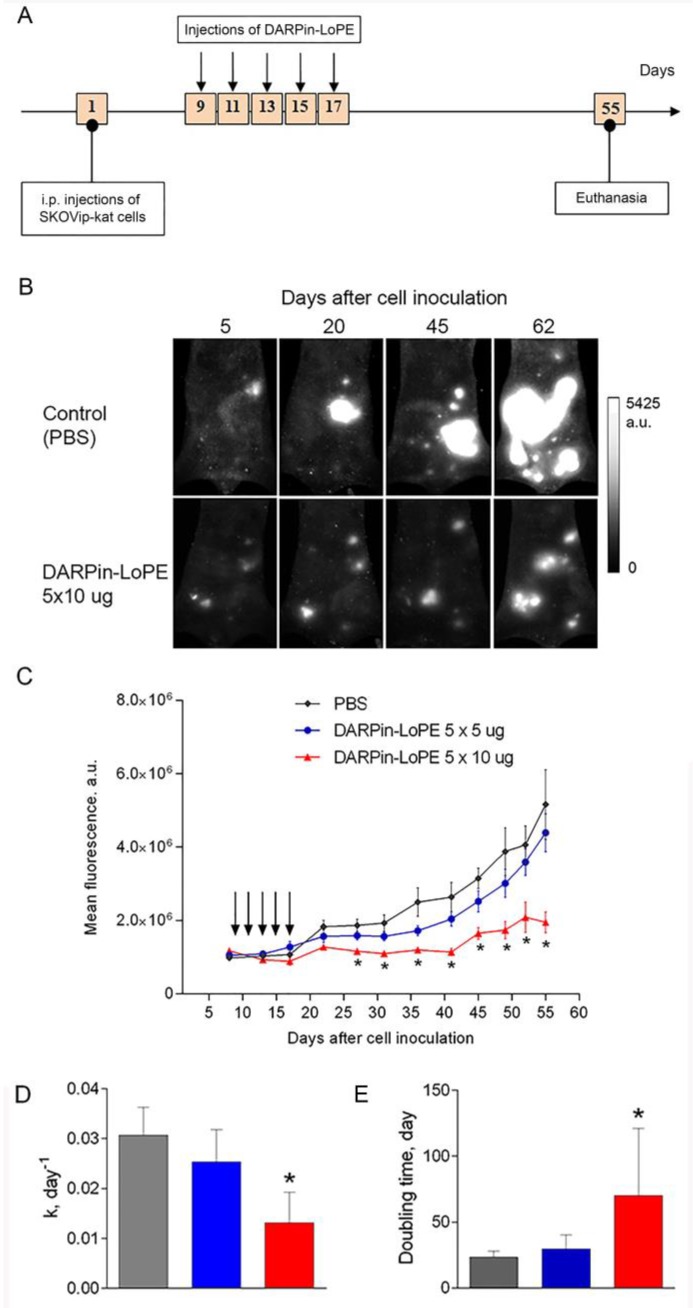
In vivo study of DARPin-LoPE against SKOVip-kat fluorescent tumor xenografts in athymic mice. (**A**) The scheme of the experiment. The day of i.p. inoculation of 4 × 10^6^ SKOVip-kat cells to animals was set as day 1; (**B**) Sequential in vivo fluorescence images of the peritoneum of control mice (injection of PBS) and mice treated with 5 × 10 µg DARPin-LoPE. In vivo 2D fluorescence images were acquired using whole-body fluorescence imaging setup with planar epi-illumination geometry. Image size 3 × 5 cm; (**C**) Tumor progression measured by in vivo fluorescence whole-body imaging in animals treated with PBS (control, black curve), 5 × 5 µg DARPin-LoPE (blue curve) or 5 × 10 µg DARPin-LoPE (red curve). The days of the targeted toxin injections are indicated with arrows; (**D**) Tumor growth rate coefficient (k) in different groups of animals; (**E**) Tumor doubling time in different groups of animals. The data are represented as mean ± SEM. “*” indicates a value that significantly differs from the respective control value at *p* < 0.05 (Dunnett’s test, *n* = 5–8).

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
