# Peer review of "HER2-Specific Targeted Toxin DARPin-LoPE: Immunogenicity and Antitumor Effect on Intraperitoneal Ovarian Cancer Xenograft Model"

_ijms, 2019, doi:10.3390/ijms20102399_

Round 1
Reviewer 1 Report
The presented manuscript by Sokolova and co-authors is devoted to the in vivo study
of the antitumor HER2-targeted toxin DARPin-LoPE composed of
non-immunoglobulin scaffold DARPin molecule and fragment of Pseudomonas
exotoxin A. The experiments are correctly designed, the data are clearly
interpreted and presented. Overall, the results of this study are
important since evident anti-tumor effect has been shown for the
targeted toxin while its general toxicity and immunogenicity are low.
This work may be accepted for publication in International Journal of
Molecular Sciences after some minor revision.
Comments:
1. Name of the targeted toxin should be revised: there are now two variants, "DARPin-LoPE" and "DARP-LoPE".
2.
Phrase "HER2 receptor" should be changed, as it doubly includes the
word "receptor". Now it is confused what do the authors meen - "HER2" or
"receptor of HER2"?
Author Response
Dear Reviewer,
We would like to express our sincere appreciation for your careful attention to our manuscript and for the suggested improvements and valuable comments. We have revised the manuscript according to your remarks. Please find below the detailed description of the revisions (Reviewer’s comments are presented in bold, our answers follow in plain text).
Comments and Suggestions for Authors:
English language and style are fine/minor spell check required
The English was carefully revised.
The presented manuscript by Sokolova and co-authors is devoted to the in vivo study of the antitumor HER2-targeted toxin DARPin-LoPE composed of non-immunoglobulin scaffold DARPin molecule and fragment of Pseudomonas exotoxin A. The experiments are correctly designed, the data are clearly interpreted and presented. Overall, the results of this study are important since evident anti-tumor effect has been shown for the targeted toxin while its general toxicity and immunogenicity are low. This work may be accepted for publication in International Journal of Molecular Sciences after some minor revision.
Comments:
1. Name of the targeted toxin should be revised: there are now two variants, "DARPin-LoPE" and "DARP-LoPE".
Corrected (the name of the targeted toxin is now “DARPin-LoPE” throughout the manuscript).
2. Phrase "HER2 receptor" should be changed, as it doubly includes the word "receptor". Now it is confused what do the authors meen - "HER2" or "receptor of HER2"?
The phrase "HER2 receptor" was replaced with "HER2 protein" or "HER2".
Reviewer 2 Report
Sokolova E and colleagues presented an interesting research article about the application of an engineered Pseudomonas aeruginosa toxin composed by a low immunogenic exotoxin epitope lacking the immunodominant human B lymphocyte domains and the HER2-specific DARPin antibody mimetic protein. In the present study, the authors tested the effects and the immunogenicity of two variants of engineered PE toxin, the new DARPin-LoPE, and the already tested DARPin-PE40, by using HER2 positive ovarian cancer cells and xenograft mice models. The authors demonstrated that the targeted toxin DARPin-LoPE showed a higher inhibitor efficacy against tumor growth in both in vitro and murine models. Furthermore, DARPin-LoPE toxin showed also lower immunogenicity compared to DARPin-PE40. Overall, the experimental design appears appropriate and the manuscript well written. Below are reported only minor comments:
1) The title of the manuscript is ambiguous because the authors refer to the immunogenicity and antitumor effect of DARPin-LoPE in vivo, however, they performed also experiments in vitro HER2-positive ovarian cancer model. It is suggested to change the title specifying the use of murine xenograft models for the in vivo experiments;
2) In the Introduction section, the authors declare “Despite a number of HER2-specific drugs approved for clinical use (trastuzumab, pertuzumab, lapatinib, trastuzumab emtansine, neratinib), successful treatment of HER2-positive tumors is still a serious problem, therefore the development of HER2-targeted drugs holds sway.”. Please provide a reference for this sentence. For this purpose see and cite:
- Falzone L, Salomone S, Libra M. Evolution of Cancer Pharmacological Treatments at the Turn of the Third Millennium. Front Pharmacol. 2018 Nov 13;9:1300. doi: 10.3389/fphar.2018.01300.
3) In the Introduction, the authors should specify for which type of tumors or disease the new proposed therapeutic approach can be used;
4) In some part of the manuscript, the English language and grammar may be improved. It is suggested English editing performed by an English native speaker;
5) In the subheading “4.3. Generation of a fluorescent human ovarian carcinoma cell line SKOVip-kat overexpressing HER2 receptor” of the Material and Methods section, please indicate how many cells were transfected and sorted;
6) In the paragraph “4.4. Confocal microscopy”, specify how many cells were seeded.
Author Response
Dear Reviewer,
First of all, we would like to express our deep gratitude for careful evaluation of our manuscript and your competent comments. We have thoroughly revised the manuscript according to your remarks. Please find below the detailed answers to the questions mentioned in your review (Reviewer’s comments are presented in bold, our answers follow in plain text).
Comments and Suggestions for Authors:
Sokolova E and colleagues presented an interesting research article about the application of an engineered Pseudomonas aeruginosa toxin composed by a low immunogenic exotoxin epitope lacking the immunodominant human B lymphocyte domains and the HER2-specific DARPin antibody mimetic protein. In the present study, the authors tested the effects and the immunogenicity of two variants of engineered PE toxin, the new DARPin-LoPE, and the already tested DARPin-PE40, by using HER2 positive ovarian cancer cells and xenograft mice models. The authors demonstrated that the targeted toxin DARPin-LoPE showed a higher inhibitor efficacy against tumor growth in both in vitro and murine models. Furthermore, DARPin-LoPE toxin showed also lower immunogenicity compared to DARPin-PE40. Overall, the experimental design appears appropriate and the manuscript well written. Below are reported only minor comments:
1) The title of the manuscript is ambiguous because the authors refer to the immunogenicity and antitumor effect of DARPin-LoPE in vivo, however, they performed also experiments in vitro HER2-positive ovarian cancer model. It is suggested to change the title specifying the use of murine xenograft models for the in vivo experiments;
We corrected the title of the manuscript in order to specify the use of xenograft models: “HER2-specific targeted toxin DARPin-LoPE: immunogenicity and antitumor effect on intraperitoneal ovarian cancer xenograft model”.
2) In the Introduction section, the authors declare “Despite a number of HER2-specific drugs approved for clinical use (trastuzumab, pertuzumab, lapatinib, trastuzumab emtansine, neratinib), successful treatment of HER2-positive tumors is still a serious problem, therefore the development of HER2-targeted drugs holds sway.”. Please provide a reference for this sentence. For this purpose see and cite:
- Falzone L, Salomone S, Libra M. Evolution of Cancer Pharmacological Treatments at the Turn of the Third Millennium. Front Pharmacol. 2018 Nov 13;9:1300. doi: 10.3389/fphar.2018.01300.
We thank the Reviewer for the valuable comment and recommended literature (it is now cited as [16]).
3) In the Introduction, the authors should specify for which type of tumors or disease the new proposed therapeutic approach can be used;
We corrected the Introduction to make this point clearer (please, see lines 76-86).
4) In some part of the manuscript, the English language and grammar may be improved. It is suggested English editing performed by an English native speaker;
The English was carefully revised.
5) In the subheading “4.3. Generation of a fluorescent human ovarian carcinoma cell line SKOVip-kat overexpressing HER2 receptor” of the Material and Methods section, please indicate how many cells were transfected and sorted;
The details were added in the paragraph 4.3 of the Materials and methods section.
6) In the paragraph “4.4. Confocal microscopy”, specify how many cells were seeded.
The details were added in the paragraph 4.4 of the Materials and methods section.